# Journal editors' perspectives on the communication practices in biomedical journals: a qualitative study

Ketevan Glonti [ID],[1,2] Isabelle Boutron,[2] David Moher [ID],[3] Darko Hren [ID][1]

[1]Department of Psychology, School of Humanities and Social Sciences, University of Split, Split, Splitsko-dalmatinska, Croatia
[2]CRESS, INSERM, INRA, Université de Paris, Paris, Île-de-France, France
[3]Ottawa Methods Centre, Ottawa Hospital Research Institute, Ottawa, Ontario, Canada

**Correspondence to**
Ketevan Glonti; kglonti@unist.hr

## ABSTRACT

**Objective** To generate an understanding of the communication practices that might influence the peer-review process in biomedical journals.

**Method** Recruitment was based on purposive maximum variation sampling. We conducted semistructured interviews. Data were analysed using thematic analysis method.

**Participants** 56 journal editors from general medicine (n=13) and specialty (n=43) biomedical journals. Most were editor-in-chiefs (n=39), men (n=40) and worked part time (n=50).

**Results** Our analysis generated four themes (1) providing minimal guidance to peer reviewers—two subthemes described the way journal editors rationalised their behaviour: (a) peer reviewers should know without guidelines how to review and (b) detailed guidance and structure might have a negative effect; (2) communication strategies of engagement with peer reviewers—two opposing strategies that journal editors employed to handle peer reviewers: (a) use of direct and personal communication to motivate peer reviewers and (b) use of indirect communication to avoid conflict; (3) concerns about impact of review model on communication—maintenance of anonymity as a means of facilitating critical and unburdened communication and minimising biases and (4) different practices in the moderation of communication between authors and peer reviewers—some journal editors actively interjected themselves into the communication chain to guide authors through peer reviewers' comments, others remained at a distance, leaving it to the authors to work through peer reviewers' comments.

**Conclusions** These journal editors' descriptions reveal several communication practices that might have a significant impact on the peer-review process. Editorial strategies to manage miscommunication are discussed. Further research on these proposed strategies and on communication practices from the point of view of authors and peer reviewers is warranted.

## INTRODUCTION

The peer-review process in biomedical journals involves collaboration between authors, journal editors and peer reviewers, which aims to achieve the dissemination of high-quality research. Good communication practices between these actors are

vital to achieve this aim. However, evidence suggests that there are numerous flaws within the peer-review process, with communication failures lying at the heart of the problem. For example, existing research suggests that an essential aspect of collaboration—the mutual understanding of stakeholders' professional roles and tasks within the process—is not appropriately communicated. This is manifested in part through the inconsistent provision of journal guidelines for peer reviewers across biomedical journals.[1] Ineffective communication practices are also manifested through the lack of transparency and considerable variation observed in the content of peer reviewers' grading forms used to evaluate original manuscripts.[2]

### Strengths and limitations of this study

► The use of in-depth qualitative interviews has provided rich data and new insights into previously hidden aspects regarding the communication practices within the peer-review process.

► We followed rigorous methodological techniques throughout the six phases of thematic analysis to ensure trustworthiness in the analysis.

► While study participants were diverse in terms of characteristics related to the journals, we were unable to include more junior editorial staff.

► There may have been social desirability bias during the interviews that affected how participants described the communication practices in their journals.

► Individuals who declined to participate in the study may have had different experiences of the peer-review process in their journals compared with those who agreed to participate.

Miscommunication typically leads to misunderstandings, which in turn might have a negative impact on different aspects of the peer-review process. For example, a study that aimed to identify tasks that journal editors expect from peer reviewers who evaluate a manuscript reporting on a

randomised controlled trial (RCT) found a substantial disconnect between the expectations of journal editors and peer reviewers. The tasks rated as important by peer reviewers were different from the tasks clearly requested by journal editors in their recommendations.[3] This can have negative impact on the quality of peer reviewers' reports as expectations on both sides remain unmet, and a delay in the publication process might arise because new peer reviewers might have to be found. Such situations can be considered to be wasteful of resources, straining the already overburdened system.[4]

Yet another study demonstrated unmet expectations caused by lack of communication that in turn influenced the willingness and motivation of peer reviewers to participate in the process. According to at least one survey, peer reviewers would like to receive feedback from journal editors about their reports and view other peer reviewers' comments, which are often not provided.[5]

The studies mentioned above directly or indirectly report on misunderstandings and miscommunication in this field. However, to the best of our knowledge, thus far there have not been any studies that specifically explore communication practices within the editorial peer-review process in biomedical journals. In order to address this gap, we set out to generate an in-depth understanding of the peer-review process with the aim of capturing social aspects that underpin or influence the process. Given that we are specifically interested in the interaction between the key actors and wanted to capture salient social and subjective dimensions of the communication, we considered a qualitative research approach to be best suited for our study aim. We therefore set out to interview journal editors. Our decision to focus on the journal editors' perspective stems from the fact that they are central figures who oversee the communication between, as well as communicate directly with, both authors and peer reviewers. Furthermore, they interact with editorial team members, publishers and readers of their journals and are therefore involved in all aspects related to communication in peer review. As indicated in the previously published study protocol,[6] this study had two complementary objectives: first, to identify journal editors' expectations and understanding regarding the roles and tasks of peer reviewers and second, to explore their perspectives and experiences of their interactions with peer reviewers and authors. The volume of rich data generated by the interviews was such that it would have been difficult to meaningfully condense the research findings into a single paper. This led to our decision to publish the findings in two distinct, yet complementary, research papers. While in the first study, we focused on editors' understanding of roles and tasks,[7] in this present study, we specifically examined how these are communicated to peer reviewers and how other interactions with peer reviewers and authors work in practice.

## METHODS
### Study design
We adopted a qualitative study design and conducted semistructured interviews with biomedical journal editors. A detailed description of the study design and methodology is available elsewhere[6] as well as in a related study using this dataset.[7] A brief description of the key methodological components follows below.

### Patient and public involvement
Patients and the public were not involved in the design, conduct, reporting or dissemination of our research.

### Sampling, recruitment and data collection
We used purposive maximum variation sampling.[8] Interviewees were recruited from multiple sources, including professional contacts, networks and directly from publishers. Eligible study participants consisted of journal editors of biomedical journals including all specialties, referring to individuals who were at the time of the interview involved in the communication process between authors and peer reviewers and/or who were in a position to decide about the fate of manuscripts. Interviewees were approached via email from multiple sources, including professional contacts; attendees of the Eighth International Congress on Peer Review and Scientific Publication, and from the *BioMed Central* and *British Medical Journal* publishing groups.

Prospective interviewees were provided with a study consent form and an information sheet. Interviewees were asked to sign a written consent form. Before starting the interview, study objectives were reiterated and additional information provided where necessary.

Since sample size is irreversibly linked to saturation, which in turn can only be operationalised during data collection,[9] our approach to data collection and analysis was iterative. Thus, recruitment continued until saturation—conceptualised as the point at which no new codes and themes were identified from the data—was achieved. All interviews were conducted by the lead author (KG) either face-to-face (n=2) or by telephone (n=2) between October 2017 and February 2018 using a topic guide (online supplementary additional file 1) and lasted 25–60 min.

While at the time of the interviews, KG was a PhD student, she has previously experienced the peer-review process in biomedical journals as an author and peer reviewer. She had training in conducting qualitative interviews prior to data collection. She was supervised by DH, who has extensive experience of the peer-review process in biomedical journals as an author, peer reviewer and journal editor.

### Analysis
Interviews were transcribed verbatim and field notes were written up after every interview. All documents were then imported into NVivo V.12 and subjected to thematic analysis as described by Braun and Clarke[10] and outlined in

**Table 1** Sample characteristics

| **Demographic characteristics** | |
|---|---|
| Sex | Female (n=16), male (n=40) |
| Position | Junior editor (n=1), senior/associate editor (n=11), coeditor-in-chief (n=4), editor-in-chief (n=39), editorial director (n=1) |
| Commitment | Part time (n=50), full time (n=6) |
| Geographic location | Asia (n=2), Africa (n=1), North America (n=19), South America (n=3), Europe (n=28), Oceania (n=3) |
| **Journal characteristics** | |
| Journal specialty | General medicine and mega journals* (n=13), specialty (n=43) |
| Indexing status† | Yes (n=53), no (n=3) |
| COPE membership‡ Peer-review model | Member (n=27), not a member (n=29) single-blind (n=38), double-blind (n=7), triple-blind (n=1), open peer review (n=9), postpublication (n=1) |
| Open access, subscription, mixed | Open access (n=35), subscription (n=4), mixed (n=17) |
| Publishers | Academic (n=9)§, commercial (n=34), mixed model¶ (n=13) |

*A peer-reviewed academic open access journal designed to be much larger than a traditional journal by exercising low selectivity among accepted articles.
†Refers to indexing status on MEDLINE, Scopus and Web of Science.
‡COPE—refers to the Committee on Publication Ethics.
§Refers to journals that are either published by universities and colleges or by independent research institutes.
¶Refers to journals that are either co-owned by medical societies and commercial publishers or owned entirely by medical societies but operated through a commercial publisher.

the protocol.[6] In summary, a preliminary codebook was generated independently by two researchers (KG and DH) from a subset of six interviews[11] using both deductive codes from topics in the interview guide and inductive content-driven codes. The remaining 50 interviews were coded by the lead researcher (KG) and supervised by DH through regular meetings. In line with the iterative process of data collection and analysis, interviews were analysed in the order in which they were conducted. To assess saturation, the lead researcher documented the process of code development, updating the codebook after analysing each transcript. Saturation was achieved after 56 interviews. To establish trustworthiness in this research, the step-by-step approach proposed by Nowell *et al*, which provides a detailed description of how to conduct a trustworthy thematic analysis, was followed.[12] This approach used criteria for trustworthiness in qualitative research proposed by Lincoln and Guba[13] and shows how these can be achieved throughout the six phases of thematic analysis. The methodological techniques that we undertook to ensure a trustworthy analysis throughout our study are presented in online supplementary additional file 2.

## RESULTS

A total of 56 biomedical journal editors were interviewed (table 1). Of these, the majority were men (n=40), editor-in-chiefs (n=39) and worked part time (n=50) at specialty journals (n=43) that employed a single-blind review process (n=38). All key characteristics of our study participants are summarised in table 1.

We identified four themes from the analysis of the interview data (1) providing minimal guidance to peer reviewers, (2) communication strategies of engagement with peer reviewers, (3) concern about impact of review model on communication and (4) different practices in the moderation of communication between authors and peer reviewers. An overview of the themes and subthemes is presented in figure 1.

### Providing minimal guidance to peer reviewers

The theme 'providing minimal guidance to peer reviewers' described the way journal editors rationalised providing peer reviewers with vague guidelines and minimal guidance around their expectations. Their perspectives in this regard coalesced around two subthemes: (a) peer reviewers should know without guidelines how to review and (b) detailed guidance and structure might have a negative effect.

### Peer reviewers should know without guidelines how to review

As a general practice across most biomedical journals, journal editors' expectations in terms of the roles and tasks of peer reviewers are communicated through the publisher's submission system. An automated email invitation typically leads peer reviewers to the online submission system for the journal, where they are presented with a 'reviewer form' interface that may include guidelines for peer reviewers to follow. Only a few journal editors,

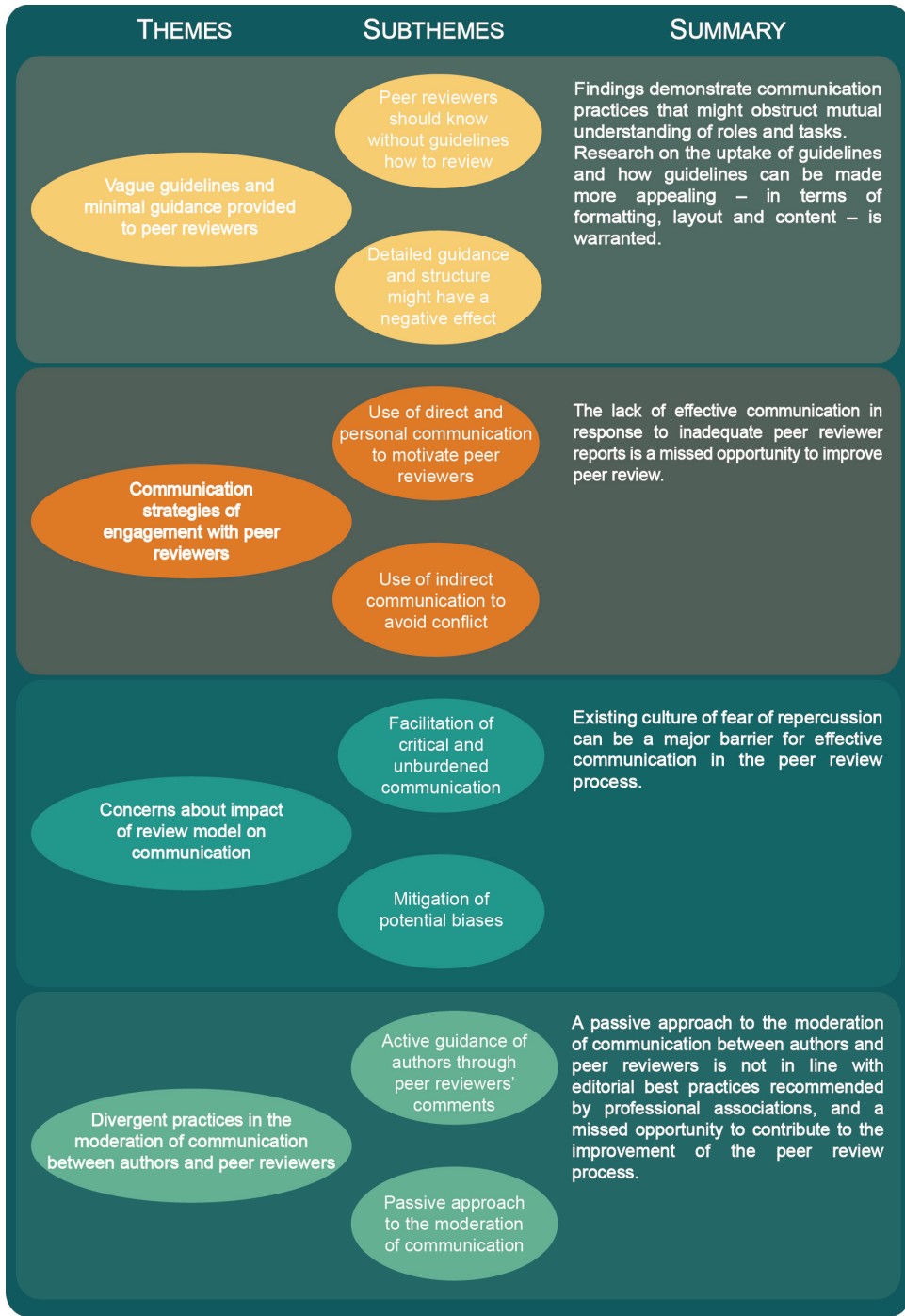

**Figure 1** Overview of the four themes.

notably those who work for non-commercial publishers, reported consciously engaging with the guidelines that are provided to peer reviewers by customising and updating them regularly. On the other hand, journal editors working with commercial publishing groups reported that publishers often 'harmonise instructions and guidelines to authors and peer reviewers across their entire range of journals' which usually results in the provision of 'standard guidelines that are meant to generally fit all types of research articles'. Thus, editors described the guidance provided by publishers to peer reviewers as 'rather vague', 'rough' and 'unspecific' in terms of concrete expectations from peer reviewers. However, this lack of specificity and detail was not considered to be an issue of concern because journal editors predominantly regarded peer reviewers' guidelines as being superfluous, indicating that peer reviewers—particularly experienced ones—are unlikely to engage with them:

> People just don't read the instructions carefully enough. If they are experienced reviewers, they are definitely not going to read the instructions. (Editor-in-chief, specialty journal)

Interviewees generally felt that written guidelines have little or no practical impact on peer reviewers' understanding of their role within the process and the quality of their performance, since such understanding is mostly dependent on the peer reviewer's experience as an author and concurrent peer-reviewing skills acquired over time:

> My feeling or my experience is that it might not matter that much what we write, because the good ones [peer reviewers] deliver good reviews anyway, and the bad ones deliver bad reviews anyway. Either you understand your role or you don't. That is at least my experience. (Editor-in-chief, general medicine journal)

There was also a prevailing assumption that detailed guidelines and specific instructions are only useful for 'inexperienced peer reviewers', whereas 'experienced peer reviewers'—described as prolific authors whose manuscripts have been reviewed, and who have reviewed manuscripts themselves, on numerous occasions—already know, or should know, what to do. Since experienced reviewers are preferred over inexperienced ones, the provision of detailed guidelines was not considered to be essential. Notably, interviewees would often speak interchangeably from both their perspective and experience as journal editors and peer reviewers in order to justify their views:

> They receive some guidance, however most of the time the reviewers who are selected are seasoned investigators themselves, and that is the reason why they are selected to review the manuscript. I have been reviewing papers for 30 years, so…we expect the reviewers to know most of the time what is involved. (Editor-in-chief, specialty journal)

### No need for detailed guidance and structure

The 'reviewer form' interface often includes boxes to fill in, checklists to complete and space for a narrative report. The reviewer forms of journals whose journal editors we interviewed varied in their structure and detail according to journal editors' preferences and the degree of customisation that was possible. To some extent, such forms prompt peer reviewers to comment on specific issues of interest and can therefore be considered an indirect form of guidance.

Most journal editors reported having a semistructured form that consists of some open questions, some closed questions and request for a narrative report. All journal editors emphasised that they place a higher premium on the 'free text' element that provides the critical insight and reflection that they seek to aid their decision-making role. Thus, the majority of interviewed journal editors expressed a preference for having a few multiple-choice questions/boxes and more space for 'free/narrative text'. Yet again, it was notable that journal editors referred to their own experiences as peer reviewers in order to support their views and justify the layout of the 'reviewer form' for their journal:

> The structured boxes, I find them kind of annoying actually when I have to fill them out for other journals. If there is too many boxes, some of these boxes become irrelevant and or I address the comments in another box and I have to put in see prior or see next box or so on, because these submission systems don't allow you sometimes to leave a box blank. So it can be annoying to the reviewer. There needs to be a happy medium between structure and free flow. (Editor-in-chief, general medicine)

Journal editors were generally open-minded and flexible with regards to the content and style of the report they expect to receive, leaving it up to peer reviewers to decide how to structure their reports. Most editors considered highly structured forms and templates that 'zoom into' the different sections of manuscripts as 'not necessarily informative' and 'not helpful' to elicit high-quality reviewers' reports. This is because excessive granularity may discourage some aspects of reviewers' 'narrative' or 'subjective opinion'—arising from their experience and expertise around the topics discussed—that editors are after. Journal editors also thought that a highly structured form could impede comments on issues that lie outside of the form's list of items. Instead, journal editors prefer to let peer reviewers' comment freely without prompt. Therefore, the majority of interviewed journal editors do not routinely share structured guidance with peer reviewers:

> We are very open and unstructured…when I reviewed for some other journals it is incredibly highly structured but not necessarily informative. I have noticed that you can have very structured peer review forms and that "makes sure" all bases are covered, but actually it is a little bit of a tick box exercise, and in our journal we simply ask reviewers to make their expert comments on all aspects of the paper that they feel they can comment on. We don't have any of that sort of tight structure, we don't ask for separate views on for instance different sections of the paper, we don't ask for separate comments about methodology. Some editors require quite excessive levels of detail on their peer review form…. we very much take the view that we want a narrative review. (Editor-in-chief, specialty journal)

Excessive structure and guidance were perceived to be prescriptive, with connotations of a 'compulsory exercise'. Instead editors felt the need to give peer reviewers a degree of autonomy and 'a feeling of freedom and creativity' to keep them motivated. This was achieved by 'non-communication' (a form of indirect communication), through not giving a structure to peer reviewers.

> They can be anyway they want them to be. When I write a review I write it in paragraphs or I might say let me talk about the intro, let me talk about the discussion, let me talk about the results all the reviews are different. But I don't think there is any problem with

a free form review, not at all. I don't want to dictate to a reviewer who is not paid to do this. This is purely voluntary so I don't want to make it an onerous task. (Coeditor-in-chief, specialty journal)

Most journal editors were also in favour of keeping the reviewer forms simple to reduce the risk of excessive 'bureaucratisation' of the process and avoid making it a 'burdensome' and 'not enjoyable' experience, which in turn could affect the willingness of peer reviewers to perform the review:

I think the risk of using templates is that… it turns the review into more of a chore, and I don't mean that it is actually correct, but just as a perception I think it might turn reviewers off. (Editor-in-chief, specialty journal)

## COMMUNICATION STRATEGIES OF ENGAGEMENT WITH PEER REVIEWERS

With the exception of editors from journals with high impact factor, most interviewees highlighted a general shortage of willing peer reviewers, so that they frequently find themselves having to act strategically to maintain their reviewing system. As part of this theme, our analysis revealed two distinct and opposite communication strategies that editors employed to handle peer reviewers: (a) *use of direct and personal communication* to motivate peer reviewers to (continuously) participate in the review process and (b) *use of indirect communication* to avoid potential conflict that could discourage peer reviewers from participating in the review process.

### Use of direct and personal communication to motivate peer reviewers

The majority of journal editors reported increasing difficulties with recruitment of peer reviewers and expressed frustration with the high decline rate, often having to contact 'numerous potential peer reviewers before finding someone who would agree to do the peer review'. Journal editors were particularly disheartened by peer reviewers who do not provide any kind of reply to invitations—'not even decline the invitation', an allegedly fairly regular occurrence. Several recruitment strategies are employed to overcome this challenge, most commonly the establishment of direct and personalised communication as opposed to the standard 'faceless' email sent through the submission system. Journal editors reported that making an effort towards a personal interaction, ideally leading to the development of a personal relationship with the reviewer, was key to establishing a 'sense of responsibility' for the reviewing task that leads to a desirable outcome:

I think one of the important points in recruiting reviewers is to contact them … the first contact is important. When you send them an e-mail or call them, it makes it easy [and] they feel a responsibility to cooperate with your journal or to help you with your

work…and fulfil the job within the time frame mentioned. (Editor-in-chief, specialty journal)

Direct communication was also used by some journal editors to preemptively increase the likelihood of receiving a high-quality review report. For example, some journal editors reported customising their communication to peer reviewers in order to draw on their expertise and call specific aspects of the manuscript to their attention. Although such personalisation was described as being time-consuming and therefore not feasible for every submission, it is considered worthwhile as it leads to high-quality reports:

I try to ask specific questions. I always say: "Any comment on this paper will be appreciated, but in particular…For example: Do you find that the western blots are valid? Do they really make the point that they say they are making?" If I have a very specific question I will ask it and I think that, that improves the quality of the review. (Editor-in-chief, specialty journal)

Direct communication was also purposefully used as a retention strategy to ensure a sustainable relationship with peer reviewers who delivered high-quality reports. Journal editors reported sending personalised positive feedback to express their gratitude. This in turn has a positive effect on the motivation and engagement of peer reviewers:

I give positive feedback but I don't give negative feedback. I just have to choose my battles…for me peer reviewers are precious resource and I think that is true with any journal but it is particularly true at our journal, for some of the reasons I mentioned earlier in our conversation there is just a much smaller pool of people who can review the papers that we receive. And so, I want to make them feel good when they have done a good job. (Editor-in-chief, specialty journal)

### Use of indirect communication to avoid conflict

Indirect communication was used more generally as a way of maintaining a working relationship with peer reviewers, irrespective of the review report quality. This was explicitly manifested in the way editors dealt with inadequate reviewers' reports. Although inadequacy in peer reviewers' reports was perceived as highly frustrating due to the delay and additional work burden generated, journal editors consistently reported a preference for indirect communication in such instances because peer reviewers were seen as 'precious resource' that 'need to be treated with care'. Thus, direct criticism/feedback on poor performance that was believed to result in a conflict, with the concomitant risk of establishing a negative relationship and losing potential reviewers altogether, is avoided. Instead, journal editors preferred to give reviewers the benefit of the doubt in the hope of receiving a better peer review in the future:

…the trouble is you never ever want to put off a peer reviewer any more than you want to put off an author because you don't know that when you next go back to them they may give you something sensible. And you definitely don't want to have it so that they will automatically decline because they have taken against you. (Editor-in-chief, specialty journal)

Therefore, journal editors preferred to 'invest time' in and establish positive relationships with 'good' peer reviewers, while generally ignoring peer reviewers who deliver inadequate reports. The most prominent strategy of dealing with poor-quality reports was to 'discard' and 'ignore' them and quickly move on to 'seeking another reviewer's report'. In some cases, non-communication (arguably a form of indirect communication) was employed to convey or express the journal editors' displeasure. For example, journal editors reported 'not even to send a thank you note' and behaving in a 'passive-aggressive' manner by recording poor performance into their submission system for future reference or excluding peer reviewers from existing reward schemes where possible:

I don't give individual feedback. However, the submission system actually is asking us to rate the review. I can rate it as very useful, not so useful, below average, which I do because they get continuing professional development points and continuing medical education points from doing the reviews…if the review was really bad, if it was really non-informative, then they won't get their points. (Editor-in-chief, specialty journal)

At the same time, journal editors' understanding of the primary goals and priorities of the peer-review process did not include improving peer reviewers' performance or educating them to write better peer reviews. Instead, their priorities are to quickly reach an editorial decision on a manuscript, thereby ensuring a fast turnaround, and to help authors to improve the manuscript:

Why we don't give feedback? We don't want to educate the reviewers…You are not trying to educate the peer reviewers, you would like to feel that the stuff that you send back to the authors is helping to educate your authors. (Editor-in-chief, specialty journal)

Despite the lack of direct feedback on reviewer performance for educational purposes, it was standard procedure across journals to send peer reviewers a copy of the decision letter sent to authors (including all reviewers' reports) 'so that they can see what the other reviewer thought of the paper and I think that is very useful feedback'. Notably, journal editors often 'hoped' that peer reviewers would read the decision letter and compare their own reports with that of other reviewers. This was considered to be an effective form of indirect feedback facilitated by journal editors. Concurrently, it is also a convenient way of indirectly offering reviewers an opportunity to learn from fellow reviewers:

We also tried to train our reviewers in an indirect way that is when a decision was completed and when we send the decision letter to the author we usually carbon copy the decision along with the comments of all the reviewers to all the reviewers so that every reviewer can see and compare their own comments with the comments of other reviewers… and that would be a form of training for them. (Editor-in-chief, specialty journal)

However, there was a degree of 'uncertainty' regarding the effectiveness of sharing the decision letters with peer reviewers. This form of indirect communication puts the onus of improving and learning on the 'interested reviewer', while obviating the need for journal editors to provide individual feedback to reviewers:

But then I don't tell bad things to reviewers; usually I tend to more often just say good things to good reviewers and then hope that some of the mediocre reviewers will just get better when they see the decision letter and how much more detailed and like expansive the comments were from one or two other reviewers. (Editor-in-chief, specialty journal)

Finally, another perceived benefit of automatically copying reviewers in the decision letter sent to authors as an educational strategy is that journal editors thereby avoid explicitly voicing their own opinion regarding the adequacy (or lack thereof) of reviewers' reports:

If we do it in that way, then later the reviewers can have a look at the other reviewer's opinion and they can learn from the other reviewer without us strongly stating that this is our opinion. (Senior/associate editor, specialty journal)

### Concerns about impact of review model on communication

This theme is centred around the preservation of anonymity as a way of facilitating angst-free communication and preventing potential biases. Most journal editors outlined why they are unwilling to commit to opening reviewer identities in their journals. Included under this theme were two interconnected categories: facilitation of critical and unburdened communication and mitigation of potential biases.

### Facilitation of critical and unburdened communication

Traditionally, many biomedical journals have employed a single-blind review model where authors are kept unaware of their peer reviewers' identity. This was also the case for the majority of journals our interviewees worked for. Journal editors were not keen to change this set-up for several reasons. Their support for maintaining peer reviewers' anonymity primarily stems from the fact that peer reviewers and authors are often either potential 'competitors for grants'; 'colleagues and/or

collaborators' or even both simultaneously. Given this situation, journal editors commented that anonymity allows peer reviewers to be 'more frank', 'more open' and 'more critical', and thus leads to 'better quality reports' than in situations where reviewers' identities are potentially known by the authors. Journal editors gave examples from their own experiences and behaviours as peer reviewers within an open peer-review process to illustrate how they tempered their true opinions to avoid causing offence that might have future negative repercussions. One editor's comment provides a good illustration of how the reviewer's communication strategy is potentially adjusted in an open review model:

> I did a peer review just recently...I think that article should have been rejected. I didn't dare suggest rejection because it was all open peer review and these were colleagues from the region, who knew me and I knew them you know and it was like 'Oh-oh. What am I going to say?' So yes, I tried to sort of be very I don't know, be as educated as possible and say maybe it is not you know, maybe it doesn't fit the article section. (Editor-in-chief, general medicine journal)

Journal editors' opinions were strengthened by their experience of peer reviewers' low uptake of the option to sign their reviews. They explained that remaining anonymous is a way for peer reviewers to ensure 'self-protection' and 'avoid potential conflict'. A number of editors also hypothesised that a lack of anonymity might negatively affect peer reviewers' review acceptance rate and curtail their ability to find peer reviewers, thus exacerbating existing recruitment difficulties:

> In a specific area like mine, you know the area is not that big, and - we have discussed this among the associate editors as well - we have never had any wish to have an open system regarding disclosure of the names of the reviewers. It would have been more difficult to find reviewers, I am quite sure. (Editor-in-chief, specialty journal)

### Mitigation of potential biases

Editors of journals who employed the single-blind or open peer-review models shared the perception that there is little to be gained by implementing a double-blind review model because 'it would be easy for everyone to figure out the identities' of reviewers, particularly in the case of small and highly specialised fields.

In contrast, while journal editors who employed a double-blind review model were aware that peer reviewers and authors might still suspect each other's identities, they felt that implementing this model remained worthwhile to prevent biases based on authorship from affecting the quality of the peer reviewers' report. For example, they referred to the potential for peer reviewers to alter their communication practices due to 'prestige bias'—where peer reviewers' judgement and objectivity are influenced by the authors' affiliation—leading to 'lenient',

poor-quality reports. Journal editors felt that anonymity helped to mitigate this type of bias:

> In my field we have the problem...let's just say there are some prestigious groups that crank out a lot of papers, of variable quality. Sometimes reviewers would see that the papers were from these famous people and they would write really short superficial reviews that were praising this work when it didn't deserve to be praised...So we just changed to blinded review and so that really solved the problem. There was a really noticeable change in fact sometimes, because I make a point of obviously sending all the decision letters to the reviewers and some people expressed interesting comments like oh my gosh I had no idea it was from this group. And some of them even went so far as to say I am glad it was blinded review...we found that whatever the case may be in the rest of the journal world for us it was better to have blinded review. (Editor-in-chief, specialty journal)

In contrast, journal editors who employ an open identity practice by default felt that it increases accountability of all parties involved, opening up the 'black box' of editorial decisions:

> I think all peer review should be open and transparent. I just think it is a better way of doing things. It is more honest to the author in that the reviewer is given their name. It is honest to readers of the papers in that for example if, if two reviewers both feel the paper should be rejected, and say so quite forthrightly within their reviews then as an editor you are not going to publish that paper with reviews that are in effect available online saying the paper should have been rejected. (Editor-in-chief, specialty journal)

### Different practices in the moderation of communication between authors and peer reviewers

In general, journal editors moderate all communication exchanges between authors and peer reviewers during the entire peer-review process. We found different practices in the way journal editors facilitated this exchange, particularly when handling peer reviewers' comments prior to forwarding them to authors. Most commonly, journal editors regarded themselves as 'curator of peer reviewers' comments'. However, the operationalisation of this role varied considerably.

### Active guidance of authors through peer reviewers' comments

Some journal editors considered 'guiding authors through peer reviewers' comments' to be one of their key tasks. This would typically happen in consensus with other editorial members. After checking the peer reviewers' reports and deciding on how to proceed with the manuscript, they then send back the peer reviewers' comments to authors with specific guidance on how to address them together with any additional editorial comments. This practice was considered to act as a 'safety net' to screen

out incorrect suggestions and provide any supplementary guidance:

> The role of the journal editor has to be to look at what comments from the reviewers are really important to improve the [manuscript] that authors should compulsorily follow. [But] others are not so important or maybe I might indeed think that there are wrong recommendations, so I have to advise the author that this is either an optional advice or even an advice that they don't have to follow. We can say to the authors 'please address explicitly the points number 1, number 3 and number 5' and in doing this we are saying to the authors 'don't worry about the points 2 and 4'. So it is not so a big problem if the reviewer is not completely right from our point of view. (Senior/associate editor, specialty journal)

### Passive approach to the moderation of communication

In contrast, other journal editors practised an alternative, less hands-on method where peer reviewers' comments are sent to authors without any editorial guidance, letting authors decide how to deal with them, including with any contradictory comments. They would then judge the comments and author replies together and make a final editorial decision. While there was some recognition that providing guidance to authors could be valuable, time constraints often prevented editors from doing so:

> Guiding authors through peer reviewer comments is something which would be certainly valuable but I have too many manuscripts to do that. It would be too much work. It is just not feasible and sometimes there are conflicting views so it is of the responsibility of the authors when they send back the revision to say "I couldn't please both reviewers, and the reason why I chose to do this revision." So I judge on that after but not before I send (it to authors). But, ideally it should be done beforehand but it is, honestly too much you know when you have so many manuscripts. (Editor-in-chief, specialty journal)

### DISCUSSION

This study is one of the first attempts to understand communication practices within the peer-review process in biomedical journals. Our findings illustrate how several communication practices that are employed in response to specific circumstances/challenges may also concurrently influence the peer-review process itself. In addition, while it is apparent that journal editors' unique threefold experience as authors, reviewers and editors inevitably shapes their attitudes and perceptions towards peer reviewing, this is likely to be both a strength and a weakness. As was evident in their responses, journal editors may unintentionally project their own experience as peer reviewers, often not evidence based, onto the entire peer-review system, potentially limiting their ability to step outside of it and critically appraise their own narrative. This can lead to attitudes and behaviours antithetical to evidence, which is ironically often a threshold to publication required by journal editors.

Many factors affect the communication between journal editors and peer reviewers. However, at the core of this interaction, certain basic principles apply. Some, such as communication of the roles and tasks that journal editors expect peer reviewers to take on and perform, might well serve as key starting points for the process. However, our study findings from the first theme indicate that journal editors do not find this transfer of information important, at least in the biomedical field. Existing literature that explores peer reviewers' guideline content and provision practices across journals showed that these are often generic, non-specific and not readily available.[1] Our study adds to this knowledge, suggesting that this vagueness is explicitly underpinned by journal editors' prevailing attitude that guidelines do not play an essential role in conveying their expectations (in terms of roles and tasks) to peer reviewers. This attitude is in line with findings that highlighted journal editors' apparent lack of appreciation for formal peer reviewer training.[7] In both cases, the justification was the same: peer reviewers should know how to review a manuscript without needing guidelines and training. Such an approach to the communication of roles and tasks is likely to be an obstacle to mutual understanding and may ultimately impact the quality of reports received. The underlying fundamental assumption is that (extensive) authorship would inevitably lead to good reviewing ability. However, thus far, there is no evidence to support this assumption[14] and further research is needed to assess whether it is actually true or peer-reviewing scientific manuscripts is a skill that can be honed through specific training. Providing guidelines to peer reviewers could be a key aspect of such training, especially because peer reviewers come from all over the world, and it is unrealistic to believe that all of them are on the same page concerning what peer-reviewing actually means. A survey of peer reviewers has shown that the most common type of peer-review 'training' comes in the form of guideline provision, most commonly journal's instructions for reviewers.[15] In the absence of formally established requirements, commonly agreed standards and widespread training programme delivery, we believe detailed guidelines to peer reviewers could be a useful starting point for editors. Given the variations observed in terms of expectations by journal editors, these would provide a common starting point and an essential reference point during the review process. Concurrently, it would be important to promote their dissemination and uptake, particularly in light of our study participants' prevailing attitudes that peer reviewers generally do not read or use guidelines at all. While it must be kept in mind that journal editors might be projecting their own behaviour when reviewing onto other peer reviewers, it is nevertheless an important point, possibly indicating that guidelines need to be presented in a better, more

appealing way. Our study data also revealed that there is a diversity of peer-review forms in terms of structure. Most journal editors preferred less structured forms and argued that it is better to let peer reviewers' comment in an unprompted manner to elicit responses that match their expertise, rather than probing for feedback on areas that they might not feel entirely confident about, but still feel obliged to fill in the relevant box. Furthermore, journal editors expressed a fear of potential bureaucratisation and 'stifling of creativity' of the process through the introduction of a rigid structure that in turn could further reduce the willingness of peer reviewers to participate in the process.

Thus, given these findings, it is evident that further research around how guidelines can be made more appealing—in terms of formatting, layout and content—is warranted. We are not aware of existing empirical evidence on peer reviewers' preferences regarding the structure of peer reviewers' forms and guidance. Research on how the uptake of guidelines and guidance among peer reviewers can be improved is also warranted. A strong evaluative component is crucial for such research to promote meaningful improvement in peer reviewers' practices. In the biomedical field, it is a well-established fact that physicians across all disciplines are resistant to adherence to clinical guidelines and there is research looking into the contextual factors around physicians and personal motivators for uptake, as well as the guidelines themselves, to understand enabling and disabling factors for uptake and implementation.[16] Thus, research on peer-review guidelines and implementation should make use of methods and knowledge gained from this field and translate it where possible accordingly.

Our results from the second theme showed that journal editors are well aware of the positive effects of direct communication and strategically use it for retention and reward purposes. This approach is in line with evidence suggesting that the establishment of personal relationships and the opportunity to network with journal editors is ranked highly among peer reviewers as a motive to participate in the peer-review process.[15] However, this study also revealed that despite consciously being aware that personal communication can be effective, it was not specifically used to improve the quality of peer reviewers' reports: journal editors would not provide direct feedback to peer reviewers who deliver inadequate peer reviewers' reports. Thus far, except for receiving an email on the final editorial decision of the reviewed manuscript and a copy of the other reviewers' reports, peer reviewers do not often receive direct meaningful feedback regarding the quality of their work. Evidence suggests that peer reviewers would like to receive feedback from a journal on their peer-review report, and this would make them more likely to accept an invitation to peer review.[15]

Additional research to assess whether this could be a missed opportunity is warranted: investing time to send peer reviewers' personalised and constructive criticism might reap dividends, whereas the current preference for indirect, impersonal communication simply perpetuates the status quo.

However, there are several barriers that might prevent the implementation of an approach that gives due importance to feedback in peer review. First, the accounts of journal editors revealed a prevalent lack of time—the vast majority of editors in biomedical journals work part time for a purely symbolic remuneration while juggling many other additional professional roles. Therefore, journal editors preferred to 'invest time' in educating authors (who in turn might become future peer reviewers for their journal), while generally ignoring peer reviewers who deliver inadequate reports. While there is evidence that peer reviewers decline review requests due to a lack of time,[17] we are not aware of studies assessing the impact of lack of time on journal editors' work. However, given that the shortage of peer reviewers is a serious and widespread issue,[17] this reluctance to educate peer reviewers is likely to be a missed opportunity. Peer reviewers review 'for free' without remuneration. A 'contractual' approach—where reviewers can expect to receive editorial comments on their reviews in lieu of formal training or instead of a fee—should be seriously considered, perhaps under a stronger inclusion of editorial board members to support journal editors with this task. In addition to potentially enhancing the quality of peer-review reports, such an approach would also increase overall review capacity.

Second, journal editors are part of the wider scholarly system; they are often researchers who compete for grants and authors who submit their manuscripts to journals. It is possible that they might fear the consequences of providing feedback. This could be perceived as unsolicited criticism of peer reviewers' work, potentially leading to conflict and far-reaching professional consequences ranging from being disadvantaged when applying to grants to unwillingness of peer reviewers to re-engage with the journal.

A third barrier is the general lack of evidence around the domain of 'quality' in peer review,[18] leaving journal editors without the tools required to methodologically assess the quality of peer reviewers' reports.

Finally, it is important to keep in mind that journal editors are not omniscient by default. For example, a study on the completeness of reporting of randomised trials published in biomedical publications highlighted that a proportion of editors did not correctly identify RCTs, suggesting that there is need for journal editors to enhance their knowledge around identification of a randomised trial and the appropriate reporting guideline (extensions) required.[19] Such examples raise questions around journal editors' training and qualifications, an area that requires further research.

Our analysis of the third theme showed that editors have diverse views on the existing peer-review models and their potential influence on communication practices in their journal. The majority of our study participants employed a peer-review model that does not display review identities by default.

They felt that maintaining anonymity would facilitate better communication practices among peer reviewers leading to high-quality reports while protecting peer reviewers from potential conflicts. This attitude reaffirms the existence of bias in the peer-review process[20] and is in line with existing research showing that survey respondents were against opening reviewer identities to authors, believing it would make review report quality worse.[21 22] This attitude was strengthened by the low uptake of peer reviewers willing to avail themselves of the option to display their identity, which was also reflected in the literature.[14] The pros and cons of blind review versus open peer review have been widely discussed, with a diversity of views and evidence suggesting that there is no one-size-fits-all solution. However, given that academia is affected by hypercompetition[23] that works on self-regulatory basis (ie, funding boards consist of researchers who evaluate other researchers' work), it could be argued that there is a deep ingrained culture of fear of repercussion—something that became evident throughout the interviews. This is a major barrier for effective communication practice and can have an impact on the quality of the review process.

The last theme revealed starkly divergent practices in the way journal editors performed their own role. In our study, this notion was exemplified by the moderation of communication between authors and peer reviewers. While some journal editors actively interjected themselves into the communication chain to guide authors through peer reviewers' comments, others prefer to remain uninvolved—forming their own opinion and decision after viewing the exchange between authors and peer reviewers. A passive approach to the moderation of communication between authors and peer reviewers is a missed opportunity to contribute to the improvement of the peer-review process and is not in line with editorial best practices recommended by professional associations. The World Association of Medical Editors stipulates that journal editors should make it clear to authors, which revisions are essential and which are optional and provide active guidance in the case of contradictory comments.[24] Some evidence shows that a system with greater editorial involvement can improve the effectiveness of peer review.[25] Evidence also suggest that at times peer reviewers are not able to pick up all methodological errors,[26] thus an active journal editor can fill in the gaps where possible. Ultimately, it is the journal editor who has overall responsibility for the manuscripts they are assigned to, therefore we believe that it is important for the journal editor to take an evidence-based approach to editorial practices and active ownership of the review process.

## Limitations

Our recruitment approach and predominant contact with editors-in-chief during the recruitment phase gave rise to a relative homogeneity of our study sample. This could have led to selection bias, which is a key limitation of this study. The limited representation of other editorial staff members typically involved in the peer-review process (such as junior editors) may limit the generalisability of our results. Therefore, there is a need to explore whether the involvement of editorial staff in other positions would have produced different and/or more nuanced findings.

## CONCLUSION

In conclusion, our study showed that there are a number of issues related to communication practices that might have a significant impact on the peer-review process and its outcomes. In the absence of effective communication among the key stakeholders, poor transfer of critical information may ultimately lead to reviewers' dissatisfaction and dissemination of low-quality research. Less visible communication failures due to embedded organisational practices and unprofessional behaviours remain a challenge. Therefore, it is important to keep the broader context in mind when attempting to enact changes the system at the organisational and individual levels. Further research into communication practices from the point of view of authors and peer reviewers will broaden our understanding of existing editorial practices and evolving communication strategies for managing miscommunication.

**Acknowledgements** The authors would like to thank Dr Sara Schroter (*BMJ*) and Dr Elizabeth Moylan (*BMC*) for providing guidance and help on the recruitment strategy of interviewees. We also would like to thank the publishers and all study participants.

**Contributors** All authors have made substantive intellectual contributions to the development of this manuscript. KG and DH jointly contributed to study conception and design. While KG led data collection, analysis and writing of the manuscript, DH led the supervision of all these steps. IB and DM have contributed to the writing of the manuscript and approved the final manuscript.

**Funding** This project was supported by the European Union's Horizon 2020 research and innovation programme under the Marie Sklodowska-Curie grant agreement number 676 207.

**Competing interests** KG and DM had an advisory role with Publons Academy. At the time of data collection for this study, KG conducted a secondment at the *BMJ*.

**Patient consent for publication** Not required.

**Ethics approval** This project has been evaluated and approved by the University of Split, Medical School Ethics Committee. Ethical approval (reference number 2181-198-03-04-17-0029) was granted in May 2017. Prospective interviewees were provided with a study consent form and a study information sheet. Interviewees were asked to sign a written consent form prior to being interviewed. Copies of the invitation letter, information sheet and consent form are available from the leading author (KG).

**Provenance and peer review** Not commissioned; externally peer reviewed.

**Data availability statement** Data are available upon reasonable request. Data may be obtained from a third party and are not publicly available. The data generated and/or analysed in the study are not publicly available due to participant anonymity, but may be available from the corresponding author on reasonable request that includes a study protocol, ethical approval and data use agreement.

**ORCID iDs**
Ketevan Glonti http://orcid.org/0000-0001-9991-7991
David Moher http://orcid.org/0000-0003-2434-4206
Darko Hren http://orcid.org/0000-0001-6465-6568

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
