## [Reviewer comments · BMJ Open]

ARTICLE DETAILS

TITLE (PROVISIONAL)	Journal editors' perspectives on the communication practices in biomedical journals: a qualitative study
AUTHORS	Glonti, Ketevan; Boutron, Isabelle; Moher, David; Hren, Darko

VERSION 1 – REVIEW

REVIEWER	Armen Yuri Gasparyan Departments of Rheumatology and Research and Development, Dudley Group NHS Foundation Trust (Teaching Trust of the University of Birmingham, UK), Russells Hall Hospital, Dudley, West Midlands, UK
REVIEW RETURNED	27-Nov-2019

GENERAL COMMENTS	The Abstract should be more specific over the methods and conclusion. As it stands, the Conclusion is vague. Introduction. Please reflect on previous publications on who are the best reviewers and how they are picked up. Specify your aim - what was the justification of interviewing the editors? Selection of editors. There may be biases related to gender and sources. How the editors were chosen? Which specialty journals were chosen? Sample characteristics. Please ascertain whether the editors were members of the ICMJE, CSE and WAME as these associations provide clear guidance for reviewers.
---

REVIEWER	Stuart T. Haines University of Mississippi School of Pharmacy, USA
REVIEW RETURNED	09-Dec-2019

GENERAL COMMENTS	Overall this is a well-written and comprehensive manuscript. The topic is relevant to biomedical journal editors and others involved in the peer review process. The investigators indicated they have adhered to the best practices standards for qualitative research. Moreover, the authors have prior experience conducting qualitative research. I believe only minor revisions are needed. The title of the manuscript is appropriate and sufficiently descriptive. The abstract includes the key/essential information regarding the objective, methods, and results. The conclusion reported in the abstract is appropriate. The introduction provides the reader with an understanding as to why this research is needed and what the authors intended to explore. Paragraph 2, Line 41 states there is a "substantial
---

	disconnect between the expectations of journal educators and peer reviewers" but does clearly state what the nature of this "disconnect" is. It should be briefly described. The methods are well described. The results are logically represented and several verbatim quotes are included to support each identified theme. The discussion comprehensively explores the results of this report in the context of previous studies. However, there were a few states where I felt the author's opinions were injected - stating how the authors believe things should be done. Some of these statements were only tangentially related to the findings in the study and aren't supported by empirical data. While these statements were clearly prefaced with "in our opinion" or "we believe", these kinds of statements aren't evidence-based. Rather than a "call" for additional research on these important questions, the authors offer their opinions on what they believe to be important and what are missed opportunities. Examples: "To the contrary, we believe that this is an outdated view. Peer reviewing scientific manuscripts is a skill that can and should be honed through specific training." " ... we believe - at a minimum - it is important to offer, clear, detailed guidelines to peer reviewers." "We believe that this is a missed opportunity: investing time to send peer reviewers personalised and constructive criticism might reap dividends, whereas the current preference for indirect, impersonal communication simply perpetuates that status quo." These opinions show up again in the manuscript's conclusion. While the abstract provides what I believe is an appropriate conclusion based on the data, the conclusion in the full text goes beyond summarizing the study findings, implications, and application to practice. "The absence of effective communication is evident in the poor transfer of critical information and the presence of miscommunication among the key stakeholders. This may ultimately lead to reviewer dissatisfaction and dissemination of low quality research." I'm not certain the data from this study can be used to support these statements. I agree the findings invite further research to test the assumptions that the journal editors expressed. But, conversely, it is not clear if the author's recommendations will lead to improved reviewer satisfaction or higher quality research. Thus, the conclusions need to be tempered. There are three items on the COREQ checklist that were not answered - items 14, 15, and 23. I would hope the authors know where the interviews were conducted and whether non-participants were present during the interviews. If not, then a brief explanation should be provided. Item 23 should be easy to answer, either the transcripts of the interviews were returned to the participants for comment or correction ... or they were not.
--	--

REVIEWER	Joeri Tijdink AmsterdamUMC, the Netherlands
REVIEW RETURNED	23-Feb-2020

GENERAL COMMENTS	Overall: Thank you for having the opportunity to review this great work. The work is complete, thorough, novel, insightful and the quality is really good. This is a journal article that you would like to review. It is thoughtful in many ways and takes a lot of things into account. There are several aspects that I find somewhat difficult and sometimes too lengthy. Below you will find some suggestions. Abstract: It looks good. The only thing I miss is the set of recommendations that are suggested in the discussion section. They are valuable and since most researchers only read abstracts of work, these can make a difference. Introduction: I can imagine that some extra ideas about the research culture and its influence on communication might be insightful P3. Line 28: is there a reference for this Methods: Which publishers were contacted? How was the interview method? Live? By phone? In writing? Was there a feedback report or 'member check'? This could be described in more detail The selection process is poorly described. Readers need more information to judge a potential selection bias. Is there a scheme how was contacted and why, inclusion criteria? I would also suggest publishing the protocol as supplementary material. Now it is hard to find. It might help to have the informed consent form and the information sheet published as a supplement P5.I5. How was the codebook updated? Is there an example? Results section: Overall, I find the results very interesting and details and consistent. I can imagine that most readers would not zoom into all these details. A suggestion could be to make it shorter to increase readability for fewer patient readers. The main 4 themes in the results do not entirely cover the content and therefore are a bit vague. Make it concrete that it entails the communication between editor and reviewer. What are divergent practices? Could you specify a bit more in the title of the theme? P6.I35. How many editors? Why them? P6.I44. Why was it not considered an issue of concern? Can you elaborate a bit more? Discussion: The discussion section is thorough and concise. However, I can imagine that it is possible to shorten it a bit to increase readability. On the other hand, the authors provide a set of recommendations on how to improve in the future and take all considerations/advantages and disadvantages into account. There are some typo's in the discussion section (p16.I21. existence?, p16 I43 word order) The list of limitations is too small. In qualitative research, there are many more limitations that can be taken into account. (ie selection bias, no generalizability, etc). Furthermore, a set of strengths is also warranted. And should be there as this study is an example of well-executed qualitative research. Finally, some considerations regarding the implication of the results
---

	can be beneficial. What are the next steps? And what would you advise to this specific journal (BMJ Open). How can they improve their peer review practices? How can they engage reviewers more? With money? With feedback? With both?
--	--

VERSION 1 – AUTHOR RESPONSE

Reviewer #1:

1. The Abstract should be more specific over the methods and conclusion. As it stands, the Conclusion is vague.

- We thank the reviewer for raising this point. We agree that our Abstract needs to be more specific. We have now updated it accordingly, keeping in mind the word limit (300 words).

Objective To generate an understanding of the communication practices that might influence the peer review process in biomedical journals.

Method Recruitment was based on purposive maximum variation sampling. We conducted semi-structured interviews. Data were analysed using thematic analysis method.

Participants 56 journal editors from general medicine (n=13) and specialty (n=43) biomedical journals. Most were editor-in-chiefs (n=39), male (n=40) and worked part-time (n=50).

Results Our analysis generated four themes: (1) Providing minimal guidance to peer reviewers – two subthemes described the way journal editors rationalised their behaviour: a) Peer reviewers should know without guidelines how to review and b) Detailed guidance and structure might have a negative effect. (2) Communication strategies of engagement with peer reviewers – two opposing strategies that journal editors employed to handle peer reviewers: a) use of direct and personal communication to motivate peer reviewers and b) use of indirect communication to avoid conflict. (3) Concerns about impact of review model on communication – maintenance of anonymity as a means of facilitating critical and unburdened communication, and minimizing biases. (4) Different practices in the moderation of communication between authors and peer reviewers – some journal editors actively interjected themselves into the communication chain to guide authors through peer reviewer comments, others remained at a distance, leaving it to the authors to work through peer reviewer comments.

Conclusions These journal editors' descriptions reveal several communication practices that might have a significant impact on the peer review process. Editorial strategies to manage miscommunication are discussed. Further research on these proposed strategies and on communication practices from the point of view of authors and peer reviewers is warranted.

2. Introduction. Please reflect on previous publications on who are the best reviewers and how they are picked up.

- We thank the reviewer for raising this point. We have presented and cited such publications in our previous, related article "*Journal editors' perspectives on the roles and tasks of peer reviewers in biomedical journals: a qualitative study*" (1) where we specifically focused on journal editors' understanding of what constitutes a good peer reviewer and on what basis they are selected or recruited. This current manuscript specifically focuses on communication practices, so adding such reflection would constitute repetition and we believe is unlikely to add substantial value to the premise of the paper without unnecessarily adding to the article's length.

3. Specify your aim - what was the justification of interviewing the editors?

- Thank you for this relevant suggestion. We agree and have now added additional detail on the aim and study objectives in the introduction section (page 3):

“The studies mentioned above directly or indirectly report on misunderstandings and miscommunication in this field. However, to the best of our knowledge, thus far there have not been any studies that specifically explore communication practices within the editorial peer-review process in biomedical journals. In order to address this gap, we set out to generate an in-depth understanding of the peer-review process with the aim of capturing social aspects that underpin or influence the process. Given that we are specifically interested in the interaction between the key actors and wanted to capture salient social and subjective dimensions of the communication, we considered a qualitative research approach to be best suited for our study aim.”

4. Selection of editors. There may be biases related to gender and sources. How the editors were chosen? Which specialty journals were chosen?

- We thank the reviewer for raising this point. We agree that it might be helpful for the reader to have more details on this key methodological aspect. We have updated the Methods section accordingly. In addition, we have included the previously published study protocol that contains a detailed description of the methodological approach as an ‘additional file’ to allow for easier access to all key details.

“We adopted a qualitative study design and conducted semi-structured interviews with biomedical journal editors. A detailed description of the study design and methodology is available elsewhere (6), as well as in a related study using this dataset (7). A brief description of the key methodological components follows below. We used purposive maximum variation sampling (8). Eligible study participants consisted of journal editors of biomedical journals including all specialties, referring to individuals who were at the time of the interview involved in the communication process between authors and peer reviewers and/or who were in a position to decide about the fate of manuscripts. Interviewees were approached via email from multiple sources, including professional contacts; attendees of the Eighth International Congress on Peer Review and Scientific Publication, and from the BioMed Central and British Medical Journal publishing groups.”

5. Sample characteristics. Please ascertain whether the editors were members of the ICMJE, CSE and WAME as these associations provide clear guidance for reviewers.

- Thank you for this suggestion. In the overview of the sample characteristics (Table 1) we provide details on membership of the Committee on Publication Ethics (COPE). Other membership details were not collected.

Reviewer #2:

1. Overall this is a well-written and comprehensive manuscript. The topic is relevant to biomedical journal editors and others involved in the peer review process. The investigators indicated they have adhered to the best practices standards for qualitative research. Moreover, the authors have prior experience conducting qualitative research. I believe only minor revisions are needed.

- We thank the reviewer for the positive feedback.

2. The title of the manuscript is appropriate and sufficiently descriptive. The abstract includes the key/essential information regarding the objective, methods, and results. The conclusion reported in the abstract is appropriate.

- We thank the reviewer for the feedback on the title and abstract. We have undertaken minor changes in the abstract based on the suggestions of the other two peer reviewers.

3. The introduction provides the reader with an understanding as to why this research is needed and what the authors intended to explore. Paragraph 2, Line 41 states there is a "substantial disconnect between the expectations of journal educators and peer reviewers" but does clearly state what the nature of this "disconnect" is. It should be briefly described.

- We agree with the reviewer and have added a sentence that briefly explains the disconnect (page 2).

"For example, a study that aimed to identify tasks that journal editors expect from peer reviewers who evaluate a manuscript reporting on a randomised controlled trial found a substantial disconnect between the expectations of journal editors and peer reviewers. The tasks rated as important by peer reviewers were different from the tasks clearly requested by journal editors in their recommendations (3)."

4. The methods are well described. The results are logically represented and several verbatim quotes are included to support each identified theme.

- We thank the reviewer for the positive feedback.

5. The discussion comprehensively explores the results of this report in the context of previous studies. However, there were a few states where I felt the author's opinions were injected - stating how the authors believe things should be done. Some of these statements were only tangentially related to the findings in the study and aren't supported by empirical data. While these statements were clearly prefaced with "in our opinion" or "we believe", these kinds of statements aren't evidence-based. Rather than a "call" for additional research on these important questions, the authors offer their opinions on what they believe to be important and what are missed opportunities. Examples: "To the contrary, we believe that this is an outdated view. Peer reviewing scientific manuscripts is a skill that can and should be honed through specific training." "... we believe - at a minimum - it is important to offer, clear, detailed guidelines to peer reviewers." "We believe that this is a missed opportunity: investing time to send peer reviewers personalised and constructive criticism might reap dividends, whereas the current preference for indirect, impersonal communication simple perpetuates that status quo". These opinions show up again in the manuscript's conclusion.

- We thank the reviewer for this comprehensive and thoughtful comment. We agree that some of our statements were not phrased appropriately in a manner that reflects the results. We have amended the discussion section in line with your suggestions (page 14-16). We have modified the statements you indicate above, rephrasing them to be less forceful and better linked to the results. Throughout the discussion section, we try to highlight what kind of research would be useful to have, and why, building upon the study findings to provide logical arguments in support of our call for research.

6. **While the abstract provides what I believe is an appropriate conclusion based on the data, the conclusion in the full text goes beyond summarizing the study findings, implications, and application to practice. "The absence of effective communication is evident in the poor transfer of critical information and the presence of miscommunication among the key stakeholders. This may ultimately lead to reviewer dissatisfaction and dissemination of low quality research." I am not certain the data from this study can be used to support these statements. I agree the findings invite further research to test the assumptions that the journal editors expressed. But, conversely, it is not clear if the author's recommendations will lead to improved reviewer satisfaction or higher quality research. Thus, the conclusions need to be tempered.**
- We thank the reviewer for this thoughtful comment. We agree that the conclusion was in need of modification. We have amended it accordingly (page 16).

"In conclusion, our study showed that there are a number of issues related to communication practices that might have a significant impact on the peer review process and its outcomes. In the absence of effective communication among the key stakeholders, poor transfer of critical information may ultimately lead to reviewer dissatisfaction and dissemination of low quality research. Less visible communication failures due to embedded organisational practices and unprofessional behaviours remain a challenge. Therefore, it is important to keep the broader context in mind when attempting to enact changes the system at the organisational and individual level. Further research into communication practices from the point of view of authors and peer reviewers will broaden our understanding of existing editorial practices and evolving communication strategies for managing miscommunication."

7. **There are three items on the COREQ checklist that were not answered - items 14, 15, and 23. I would hope the authors know where the interviews were conducted and whether nonparticipants were present during the interviews. If not, then a brief explanation should be provided. Item 23 should be easy to answer, either the transcripts of the interviews were returned to the participants for comment or correction ... or they were not.**
- Thank you for pointing out this lack of clarity. We have now made an explicit note of these points in the updated reporting guideline.

Reviewer #3:

1. **Thank you for having the opportunity to review this great work. The work is complete, thorough, novel, insightful and the quality is really good. This is a journal article that you would like to review. It is thoughtful in many ways and takes a lot of things into account. There are several aspects that I find somewhat difficult and sometimes too lengthy. Below you will find some suggestions.**
- We thank the reviewer for the encouraging feedback.
2. **Abstract: It looks good. The only thing I miss is the set of recommendations that are suggested in the discussion section. They are valuable and since most researchers only read abstracts of work, these can make a difference.**

- We thank the reviewer for raising this point. We have now updated it accordingly, keeping in mind the word limit (300 words).

Objective To generate an understanding of the communication practices that might influence the peer review process in biomedical journals.

Method Recruitment was based on purposive maximum variation sampling. We conducted semi-structured interviews. Data were analysed using thematic analysis method.

Participants 56 journal editors from general medicine (n=13) and specialty (n=43) biomedical journals. Most were editor-in-chiefs (n=39), male (n=40) and worked part-time (n=50).

Results Our analysis generated four themes: (1) Providing minimal guidance to peer reviewers – two subthemes described the way journal editors rationalised their behaviour: a) Peer reviewers should know without guidelines how to review and b) Detailed guidance and structure might have a negative effect. (2) Communication strategies of engagement with peer reviewers – two opposing strategies that journal editors employed to handle peer reviewers: a) use of direct and personal communication to motivate peer reviewers and b) use of indirect communication to avoid conflict. (3) Concerns about impact of review model on communication – maintenance of anonymity as a means of facilitating critical and unburdened communication, and minimizing biases. (4) Different practices in the moderation of communication between authors and peer reviewers – some journal editors actively interjected themselves into the communication chain to guide authors through peer reviewer comments, others remained at a distance, leaving it to the authors to work through peer reviewer comments.

Conclusions These journal editors' descriptions reveal several communication practices that might have a significant impact on the peer review process. Editorial strategies to manage miscommunication are discussed. Further research on these proposed strategies and on communication practices from the point of view of authors and peer reviewers is warranted.

Introduction

3. I can imagine that some extra ideas about the research culture and its influence on communication might be insightful.

- Thank you for this suggestion. While this is an interesting suggestion, we believe that it is not directly relevant and might detract from the focus of the article.

4. P3. Line 28: is there a reference for this

- Thank you for pointing this out. We have added a reference as suggested.

Methods

1. Which publishers were contacted? The selection process is poorly described. Readers need more information to judge a potential selection bias. Is there a scheme how was contacted and why, inclusion criteria?

- We thank the reviewer for raising this point. We agree that this information should be provided in the manuscript. We have updated the Methods section accordingly. In addition, we have included the previously published study protocol as an 'additional file' to the manuscript. The protocol contains a detailed description of which publishers were contacted as well as other useful information.

"We adopted a qualitative study design and conducted semi-structured interviews with biomedical journal editors. A detailed description of the study design and methodology is available elsewhere (6), as well as in a related study using this dataset (7). A brief description of the key methodological components follows below. We used purposive maximum variation sampling (8). Eligible study participants consisted of journal editors of biomedical journals including all specialties, referring to

individuals who were at the time of the interview involved in the communication process between authors and peer reviewers and/or who were in a position to decide about the fate of manuscripts. Interviewees were approached via email from multiple sources, including professional contacts; attendees of the Eighth International Congress on Peer Review and Scientific Publication, and from the BioMed Central and British Medical Journal publishing groups.”

2. How was the interview method? Live? By phone? In writing?

- Thank you for this comment. This information was already included in the manuscript. However, based on your suggestion we have now specified the number of interviews conducted face-to-face and by phone respectively.

“All interviews were conducted by the lead author (KG) either face-to-face (n=2) or by telephone (n=54) between October 2017 and February 2018 using a topic guide (Additional file 1) and lasted 25–60 minutes.”

3. Was there a feedback report or ‘member check’? This could be described in more detail.

- This is an interesting and contentious point. Participant checking was not performed. One of the reasons for not doing was because journal editors are very busy and overburdened and they barely made time for the interview so additional checking was not considered as feasible.

It was nor was it considered a key aspect of ensuring trustworthiness of our methods and results. In qualitative literature, it is not considered a methodological ‘best-practice’. We believe that this stance is amply supported by available literature around qualitative methodology (for example: Peditto K. Reporting Qualitative Research: Standards, Challenges, and Implications for Health Design. *HERD: Health Environments Research & Design Journal*. 2018 Apr;11(2):16-9).

Instead, we have used other means to ensure that important concepts have not been misinterpreted, as outlined in detail in the additional file “Trustworthiness of analysis”. For example we used multiple returns to raw data to check for referential adequacy by the research team.

Based on your suggestion we have now explicitly noted that participant checking was not performed in the reporting guidelines checklist.

4. I would also suggest publishing the protocol as supplementary material. Now it is hard to find.

- We agree with this suggestion and have included the protocol as part of the supplementary material.

5. It might help to have the informed consent form and the information sheet published as a supplement.

- Thank you for this comment. The informed consent describes details on data storing and issues related to GDPR. We do not see any additional value in publishing this. Similarly, the information sheet summarizes study aims and objectives of the proposed study, and does not contain any information that is not already presented in this manuscript.

6. How was the codebook updated? Is there an example?

- Thank you for this comment. As described in the Analysis section below, the codebook was updated after each interview, i.e. this an iterative process whereby codes are continuously added, amended or removed as need data is captured with each interview. Figure 1, which presents the themes and subthemes, represents the last version of our codebook where the codes have been collapsed into themes and subthemes.

“In summary, a preliminary codebook was generated independently by two researchers (KG and DH) from a subset of six interviews (11) using both, deductive codes from topics in the interview guide and inductive content-driven codes. The remaining 50 interviews were coded by the lead researcher (KG), supervised by DH through regular meetings. In line with the iterative process of data collection and analysis, interviews were analysed in the order in which they were conducted.”

Results

7. **Overall, I find the results very interesting and details and consistent. I can imagine that most readers would not zoom into all these details. A suggestion could be to make it shorter to increase readability for fewer patient readers.**

- Thank you for this comment. Although we understand your concern, we feel that rich context supported by verbatim quotes is the most appropriate way to present qualitative data. We have amended the manuscript several times and feel that we have condensed our findings in such a way that readers who are not necessarily experts on this topic can fully understand how to place the results within the broader context.

The main 4 themes in the results do not entirely cover the content and therefore are a bit vague. Make it concrete that it entails the communication between editor and reviewer.

- Thank you for this comment. In the light of your comment, we have now reviewed the theme names and descriptions and have modified them for a better understanding. In addition each of the four theme titles are followed by a short summative paragraph that explains their meaning and the underlying subthemes. The latter are the key headings for the content analysis. Given the nature of qualitative research, we believe that the four main themes with associated sub-themes is an effective way of presenting our rich results in ‘bite-sized’ chunks that readers can understand. Here is the overview of the four themes:

Providing minimal guidance to peer reviewers

The theme “Providing minimal guidance to peer reviewers” described the way journal editors rationalised providing peer reviewers with vague guidelines and minimal guidance around their expectations. Their perspectives in this regard coalesced around two subthemes: a) Peer reviewers should know without guidelines how to review b) Detailed guidance and structure might have a negative effect.

Communication strategies of engagement with peer reviewers

With the exception of editors from journals with high impact factor, most interviewees highlighted a general shortage of willing peer reviewers so that they frequently find themselves having to act strategically to maintain their reviewing system. As part of this theme, our analysis revealed two distinct and opposite communication strategies that editors employed to handle peer reviewers: a) *Use of direct and personal communication* to motivate peer reviewers to (continuously) participate in the review process; and b) *Use of indirect communication* to avoid potential conflict that could discourage peer reviewers from participating in the review process.

Concerns about impact of review model on communication

This theme is centred around the preservation of anonymity as a way of facilitating angst-free communication and preventing potential biases. Most journal editors outlined why they are unwilling to commit to opening reviewer identities in their journals. Included under this theme were two interconnected categories: facilitation of critical and unburdened communication and mitigation of potential biases.

Different practices in the moderation of communication between authors and peer reviewers

Generally, journal editors moderate all communication exchanges between authors and peer reviewers during the entire peer review process. We found different practices in the way journal editors facilitated this exchange, particularly when handling peer reviewer comments' prior to forwarding them to authors. Most commonly, journal editors regarded themselves as "curator of peer reviewer comments". However, the operationalisation of this role varied considerably.

What are divergent practices? Could you specify a bit more in the title of the theme?

- We agree that the word 'divergent' might not be clear and have replaced it with 'different'. The theme is otherwise described in further detail in the summative paragraph below:

"Different practices in the moderation of communication between authors and peer reviewers"

Generally, journal editors moderate all communication exchanges between authors and peer reviewers during the entire peer review process. We found different practices in the way journal editors facilitated this exchange, particularly when handling peer reviewer comments' prior to forwarding them to authors. Most commonly, journal editors regarded themselves as "curator of peer reviewer comments". However, the operationalisation of this role varied considerably."

8. P6.I35. How many editors? Why them?

- After going through the indicated page and line number, we remain uncertain as to how we can respond to this comment. We stated that all journal editors emphasized this aspect: "***All** journal editors emphasized that they place a higher premium on the 'free text' element that provides the critical insight and reflection that they seek to aid their decision-making role.*"

9. P5.I44. Why was it not considered an issue of concern? Can you elaborate a bit more?

- Thank you for this comment. We think that this issue is elaborated upon in the following sentence and is supported by the quote. We have now combined the two sentences as follows to make this link clear.

"However, this lack of specificity and detail was not considered to be an issue of concern because journal editors predominantly regarded peer reviewer guidelines as being superfluous, indicating that peer reviewers - particularly experienced ones - are unlikely to engage with them."

Discussion

- 10. The discussion section is thorough and concise. However, I can imagine that it is possible to shorten it a bit to increase readability. On the other hand, the authors provide a set of recommendations on how to improve in the future and take all considerations/advantages and disadvantages into account.**

- Thank you for this comment. We have updated the discussion in the accordance with comments from the second peer reviewer.

11. There are some typo's in the discussion section (p16.l21. existence?, p16 l43 word order).

- Thank you for pointing out these typos. We have now corrected them.

12. The list of limitations is too small. In qualitative research, there are many more limitations that can be taken into account. (ie selection bias, no generalizability, etc). Furthermore, a set of strengths is also warranted. And should be there as this study is an example of well-executed qualitative research.

- Thank you for your comment. We have described both selection bias and generalizability in our current limitation section. We have now updated this section to make it more explicit (see below). In addition, the editor asked us to revise the 'Strengths and limitations' that follows after the abstract, that relate specifically to the methods. We believe that these two sections provide a comprehensive overview of our strengths and limitations.

'Strengths and limitations'

- The use of in-depth qualitative interviews has provided rich data and new insights into previously hidden aspects regarding the communication practices within the peer review process
- We followed rigorous methodological techniques throughout the six phases of thematic analysis to ensure a trustworthy analysis
- While study participants were diverse in terms of characteristics related to the journals, we were unable to include more junior editorial staff
- There may have been social desirability bias during the interviews that affected how participants described the communication practices in their journals
- Individuals who declined to participate in the study may have had different experiences of the peer review process in their journals compared with those who agreed to participate

“Our recruitment approach and predominant contact with editors-in-chief during the recruitment phase gave rise to a relative homogeneity of our study sample. This could have led to selection bias, which is a key limitation of this study. The limited representation of other editorial staff members typically involved in the peer review process (such as junior editors) may limit the generalizability of our results. Therefore, there is a need to explore whether the involvement of editorial staff in other positions would have produced different and/or more nuanced findings.”

13. Finally, some considerations regarding the implication of the results can be beneficial. What are the next steps? And what would you advise to this specific journal (BMJ Open). How can they improve their peer review practices? How can they engage reviewers more? With money? With feedback? With both?

- Thank you for this interesting comment. We have provided a set of recommendations on the implication of the results and how editorial practices can potentially be improved. We believe that your suggestions go beyond the scope of this manuscript.

References

1. Glonti K, Boutron I, Moher D, Hren D. Journal editors' perspectives on the roles and tasks of peer reviewers in biomedical journals: a qualitative study. *BMJ Open*. 2019;

VERSION 2 – REVIEW

REVIEWER	Stuart T Haines University of Mississippi School of Pharmacy Jackson, MS USA
REVIEW RETURNED	01-May-2020

GENERAL COMMENTS	The authors are to be applauded for making substantial revisions to this manuscript in response to peer review comments. The report is well-written, the methods and results are clearly stated, and the conclusions are supported by the qualitative analysis. While some of the recommendations the authors suggest are not evidence-based, they make clear that these are merely suggestions based on their own experiences or best-practices recommended by professional societies.
---

REVIEWER	Joeri Tijdink AmsterdamUMC, the Netherlands
REVIEW RETURNED	23-May-2020

GENERAL COMMENTS	Thanks for improving the manuscript.
--------------------------------------